# Apigenin and Hesperidin Downregulate DNA Repair Genes in MCF-7 Breast Cancer Cells and Augment Doxorubicin Toxicity

**DOI:** 10.3390/molecules25194421

**Published:** 2020-09-26

**Authors:** Agnieszka Korga-Plewko, Monika Michalczyk, Grzegorz Adamczuk, Ewelina Humeniuk, Marta Ostrowska-Lesko, Aleksandra Jozefczyk, Magdalena Iwan, Marta Wojcik, Jaroslaw Dudka

**Affiliations:** 1Independent Medical Biology Unit, Medical University of Lublin, 8b Jaczewski Street, 20-090 Lublin, Poland; monikam0327@gmail.com (M.M.); grzegorzadamczuk@umlub.pl (G.A.); ewelina.humeniuk@vp.pl (E.H.); 2Department of Toxicology, Medical University of Lublin, 6 Chodzko Street, 20-093 Lublin, Poland; martaostrowskalesko@umlub.pl (M.O.-L.); magdalena.iwan@umlub.pl (M.I.); jaroslaw.dudka@umlub.pl (J.D.); 3Department of Pharmacognosy with Medicinal Plant Laboratory, Medical University of Lublin, 1 Chodzko Street, 20-090 Lublin, Poland; ajozefczyk@pharmacognosy.org; 4Department of Pathophysiology, University of Life Sciences, 12 Akademicka Street, 20-033 Lublin, Poland; marta.wojcik@up.lublin.pl

**Keywords:** apigenin, doxorubicin, hesperidin, DNA repair, DNA damage, oxidative stress

## Abstract

A number of studies have confirmed anti-tumor activity of flavonoids and their ability to enhance the effectiveness of classical anticancer drugs. The mechanism of this phenomenon is difficult to explain because of the ambivalent nature of these compounds. Many therapeutic properties of these compounds are attributed to their antioxidant activity; however, it is known that they can act as oxidants. The aim of this study was to assess the influence of apigenin and hesperidin on MCF-7 breast cancer cells with doxorubicin. The cytotoxic effect was determined using an MTT test and cell cycle analysis. To evaluate the possible interaction mechanism, reduced glutathione levels, as well as the DNA oxidative damage and the double strand breaks, were evaluated. Additionally, mRNA expression of genes related to DNA repair was assessed. It was demonstrated that flavonoids intensified the cytotoxic effect of doxorubicin despite flavonoids reduced oxidative damage caused by the drug. At the same time, the number of double strand breaks significantly increased and expression of tested genes was downregulated. In conclusion, both apigenin and hesperidin enhance the cytotoxic effects of doxorubicin on breast cancer cells, and this phenomenon occurs regardless of oxidative stress but is accompanied by disorders of DNA damage response mechanisms.

## 1. Introduction

Considering the growing number of cancer patients and the frequent ineffectiveness of therapies, the priority seems to be searching new and more effective treatments using natural compounds with strong biological activity such as flavonoids—plant compounds that constitute an important element in the human diet [1]. A considerable amount of studies have established their therapeutic properties: anti-inflammatory, antiviral, antiallergic, diuretic, detoxifying, anti-arrhythmic, antihypertensive and anti-cancer [2]. The therapeutic properties of flavonoids are largely attributable to their antioxidant properties. The antioxidant activity of individual flavonoids is contingent on the number of hydroxyl groups and their location. It has been proved that the more hydroxyl groups in the molecule, the stronger the antioxidant effect. These properties are also enhanced by the locus of these groups in para and ortho positions [3]. However, there is an increasing number of research studies, proving the antitumor effect of flavonoids is possible not only owing to their antioxidant properties, but also due to other mechanisms connected with the inhibition of cell proliferation, angiogenesis, induction of apoptosis or reduction in protein tyrosine kinases activity [4]. What is more, some flavonoids, depending on the concentration and partial pressure of oxygen in the cells, may have a pro-oxidative effect and in this way kill cancer cells [5]. In addition to their cytotoxic properties to cancer cells, many studies have shown that these compounds enhance the effects of standard chemotherapeutics [6,7,8].

Our team is particularly interested in apigenin and hesperidin—their anticancer properties and the mechanism of sensitization of cancer cells to standard chemotherapeutics. Similar to other flavonoid compounds, they can ambivalently affect DNA. It has been established that flavonoids, owing to their antioxidant properties, have the ability to protect DNA against free radicals, preventing mutations in proto-oncogens, rebuilding genomic stability, thereby limiting the cancerous process. It was evidenced that the antioxidant activity of hesperidin was not only attributed to its radical scavenging activity, but it also boosts the cell’s antioxidant defense mechanisms [9,10,11]. Studies with in vivo models have shown that hesperidin inhibits tumorigenesis through its anti-inflammatory effects [12].

On the other hand, it has been proven that these natural compounds might trigger DNA damage through different mechanisms, i.e., through inhibiting DNA synthesis by suppressing the enzymes involved in the replication of DNA (polymerase II, topoisomerase I and II), oxidative damage or DNA intercalation [13]. It was revealed that apigenin induces DNA damage causing downregulation of genes involved in cell cycle control and DNA repair [14]. Consequently, cancer cells may be unable to repair DNA, leading to pro-apoptotic effects. There are no reports of direct interaction of hesperidin with DNA, but there are a series of reports that hesperidin acts on cancer cells, including breast, liver, esophagus and stomach cancer by triggering the accumulation of radical species [15,16,17,18]. The studies of Wang revealed that apigenin was able to inhibit estradiol-induced DNA synthesis in human breast cancer cells [19,20]. Affecting DNA structure and repair mechanisms, flavonoids can enhance radiotherapy and chemotherapy. Apigenin has shown synergistic effects with 5-fluorouracyl, paclitaxel and doxorubicin, among others [21,22,23]. It has also been shown in vitro studies that apigenin has enhanced radio-sensitivity of human lung cancer cells [24]. Hesperidin has shown synergistic effects in combination with doxorubicin, cyterbine and tamoxifen [25,26,27]. In vivo, both apigenin and hesperidin (in the complex with diosmin) revealed a radiosensitizing effect in mouse Ehrlich carcinoma [28,29]. However, there is still insufficient knowledge regarding the relationship between flavonoids and the response to DNA defects. Breast cancer is an area where both apigenin and hesperidin can find therapeutic applications. It is very important that the cytotoxic action of these flavonoids on cancer cells is accompanied by a lack of toxicity to healthy cells [30,31,32]. Moreover, both flavonoids act synergistically with the doxorubicin—a chemotherapeutic agent used to treat breast cancer. Doxorubicin is an effective anti-cancer drug that has been used for many years. This drug intercalates into DNA to inhibit topoisomerase II and leads to obstruct DNA replication and transcription [33]. The second mechanism of action is related to the generation of oxidative stress, both in cancer and healthy cells [34,35]. Synergistic effects of doxorubicin and certain flavonoids, including apigenin and hesperidin, have been repeatedly demonstrated in different types of cancer cells [23,36,37,38,39]. However, the exact mechanism of this phenomenon has not been explained. 

The aim of this study was to assess the influence of apigenin and hesperidin on MCF-7 breast cancer cells treated with doxorubicin, including DNA damage and DNA repair gene expression to better understand observed interactions.

## 2. Results

### 2.1. Cytotoxicity Analyses

The MTT test confirmed toxicity of 1 μm doxorubicin (DOX) against MCF-7 on the level of IC50 value (52.8 ± 3.59% viability). The concentration of flavonoids was chosen on the basis of our previous preliminary studies; results are shown in Table 1 where 50 µM of apigenin and 50 µM of hesperidin were optimal to sensitize the cells on DOX treatment. Apigenin alone had a similar cytotoxic effect as DOX—49.02 ± 4.23%. Hesperidin treatment revealed no cytotoxic effect. Simultaneous treatment with DOX and flavonoids showed a synergistic effect on MCF-7 cells—15.35 ± 1.75% residual viability for DOX + API (apigenin) and 19.93 ± 2.89% for DOX + HESP (hesperidin) (Figure 1).

The MTT test results are consistent with cells’ morphology analysis. The control cells showed normal, epithelial-like morphology. After treatment with DOX, API, DOX + API and DOX + HESP the cells became round, shrunk and had poor adhesion. These features were the most intense in cultures simultaneously treated with combination of DOX and flavonoid. The cells treated with hesperidin alone were not different from control cells (Figure 2).

### 2.2. Cell Cycle Analysis

Cell cycle analysis revealed that DOX and apigenin present similar histogram patterns—both compounds significantly reduced the population of cells in G1 phase, while the percentage of cells in the subG1 phase was elevated. The strongest effect can be observed with the simultaneous treatment with doxorubicin and flavonoids—the percentage of dead cells reached almost 100%. Hesperidin alone has no effect on the cell cycle—the obtained results were similar to the control sample (Figure 3).

### 2.3. Apoptosis Detection

Image cytometry analysis revealed that cell death observed in subG1 phase of cell cycle was clearly apoptotic death. After DOX treatment, the cells were in the early stages of apoptosis. In the case of simultaneous DOX and flavonoid, the number of cells in the late phase of apoptosis increased. The similar pattern has been observed in the case of API. Obtained histograms confirmed that hesperidin in tested concentration is not cytotoxic for the MCF-7 cell line (Figure 4).

### 2.4. Determination of DNA Oxidative Damage

The determination of oxidative DNA damage evidenced that DOX and hesperidin alone showed the strongest effect on the accumulation of AP sites in DNA isolated from MCF-7 cells (2.76 ± 0.11, 1.88 ± 0.28 AP sites/100 k bp, respectively), compared to the control culture (0.83 ± 0.17 AP sites/100 kbp). However, after combining DOX with hesperidin or apigenin, a decrease in the level of AP sites was observed (0.31 ± 0.04 and 0.57 ± 0.15/100 kbp, respectively). There was no significant change after treatment with apigenin (1.06 ± 0.03/100 kbp) in comparison to control (Figure 5).

### 2.5. DNA Double-Strand Breaks (DSBs)

DNA double-strand breaks (DSBs) were measured by detection based on antibodies against phosphorylated H2AX in the cell nucleus. The H2AX level was significantly higher after treating cells with DOX and apigenin alone (191.43 ± 35.24 and 131.71 ± 118.42% of control, respectively). However, a crucial rise in phosphorylated H2AX in the nucleus was noticed after combining DOX with apigenin or hesperidin (282.91 ± 29.65 and 277.11 ± 26.35% of control). Treating cells with hesperidin had no significant impact on the H2AX phosphorylation in the nucleus as compared to the control (102.13 ± 8.60% of control, Figure 6).

### 2.6. Reduced Glutathione Level

All of the test compounds, DOX as well as flavonoids, caused a reduction in GSH at a similar level (61.33 ± 5.85, 74.20 ± 6.47 and 64.73 ± 7.17% of GSH in control cells for DOX, API and HESP, respectively). Unexpectedly in the cells treated simultaneously with DOX and one of the flavonoids. The GSH level did not differ from the level in the control cells (Figure 7).

### 2.7. The Quantitative Real-Time PCR Analysis (qRTt-PCR)

Relative gene expression assessment revealed that all tested compounds (DOX, apigenin and hesperidin) caused downregulation of genes connected to DNA repair. Only in the case of PARP1, DOX did not change this gene expression level compared to the control. XPC was the only gene that was upregulated by DOX. Simultaneous treatment with DOX and apigenin caused significant downregulation of PARP1, ERCC1, OGG1, MGMT, XPC and MLH1 in comparison to single compounds. In the cases of DOX and hesperidin treatment downregulation vs. single compounds was observed for PARP1, ERCC1, MSH2, OGG1, MGMT and MLH1 (Table 2).

## 3. Discussion

A number of studies have confirmed that the anti-tumor activity of flavonoids results from their ability to induce the process of apoptosis or cell cycle arrest [40]. Genotoxic effects were associated with pro-oxidative activity. On the other hand, previous studies have concluded that ROS are not involved in apigenin-induced DNA damage, but they depend on the activation of caspases or inhibition of the enzymes necessary for the replication process [5,41]. Other sources suggest that flavonoids may intercalate into the DNA strand and might be averse to topoisomerase activity [13]. Apigenin has been established to induce DNA damage by downregulating genes concerned with cell cycle control and DNA repair [14]. Conversely these natural substances, due to their antioxidant properties, protect against DNA damage caused by various carcinogens, including reactive oxygen species [42]. Consequently, they limit DNA damage, prevent the genetic instability that underlies the formation of tumors. It has been investigated that the protective effect of selected flavonoids on DNA occurred by reducing oxidative damage [43,44,45]. Thus, they also have the ability to alleviate side effects of chemotherapy. However, the antioxidant effect of these substances might be, at the same time, unfavorable in treatment because they can weaken the effectiveness of drugs that act by generating free radicals [46]. It is difficult to predict the exact path of flavonoid activity. The possibility of using these compounds in therapy depends on understanding the exact mechanism of action.

The above observations suggest that flavonoids may have a similar mechanism of action as the known mechanisms of doxorubicin. Anticancer activity of this drug is connected with inhibition of topoisomerase II, polymerases of RNA and DNA, helicases and enzymes that repair DNA damage. Additionally, doxorubicin intercalates to DNA and prevents synthesis of RNA and DNA [47]. The second mechanism is connected with generation of the reactive oxygen species (ROS) by the drug which causes cell death in both cancer and normal cells [34,35]. Unfortunately, among cancer patients, multi-drug resistance appears increasingly. However, anti-tumor actions of flavonoids aid in the increase in the effectiveness of chemotherapy [8]. There have been several reports of sensitization of cancer cells to doxorubicin by apigenin and hesperidin [23,36,37,38,39].

In the present study, MCF-7 cells were incubated in medium supplemented with doxorubicin and one of flavonoids—apigenin and hesperidin or combined. Apigenin and hesperidin were used in optimal concentrations that sensitized the cells on DOX treatment. Synergistic effect was confirmed by an MTT test, morphological assessment and cell cycle analysis. Surprisingly, the cell cycle inhibition was not observed. In the case of DOX, API, DOX + API and DOX + HESP, we observed an increase of cells in subG1 phase that corresponds to the population of dead cells. The DNA damage analysis revealed ambiguous results. 

As DNA damage generated by free radicals plays a crucial role in the mechanism of DOX action, oxidative DNA damages have been determined using AP site measurement. DOX, as well as hesperidin, has been found to significantly increase DNA oxidative damage. However, in combination, there is a considerable reduction in the level of AP sites in the genetic material. This clearly demonstrates antioxidant properties of apigenin and hesperidin co-administered with a chemotherapeutic. The observed phenomenon may indicate an antioxidant, protective effect of flavonoids combined with a chemotherapeutic [48]. These observations are confirmed by the results of GSH level evaluation–a low molecular weight antioxidant which, under oxidative stress conditions, is converted into an oxidized form of GSSG. The antioxidant activity of the combination of doxorubicin and hesperidin/apigenin was unexpected as single agents (with the exception of single apigenin) showed prooxidant effects. Similar results were obtained by our team in the study of HepG2 cell [39]. The prooxidant effect might be a consequence of the ability of these substances to oxidize reduced form of nicotinamide adenine dinucleotide NADH upon production of phenoxyl radicals in the presence of cellular peroxidases and hydrogen peroxide [49,50]. Rusak et al. [51] analyzed the influence of apigenin on DNA of hydrogen peroxide stressed human peripheral lymphocytes and stated that the balance between the protection of DNA from oxidative damage and prooxidative effects was strongly dependent on flavonoid concentration and the incubation period. Given the fact that both flavonoids and doxorubicin undergo redox cycling, there may be some kind of interaction between radical forms of these compounds. Nevertheless, apigenin and hesperidin intensified the cytotoxic effect of doxorubicin, which means that oxidative stress is not a key factor for observed phenomenon. 

DSBs are the most serious DNA damage. They can be caused by exposure to radiation or chemicals. DSB formation was observed in cells treated with DOX and apigenin, whereas in cells treated simultaneously with DOX and flavonoid (apigenin as well as hesperidin) a significant intensity of this phenomenon was observed. Taking into account the simultaneous lack of inhibition of the cell cycle in these cells, attention was paid to the DNA damage response (DDR). DDR is a complex process that detects and repairs DNA errors to prevent cancer using elaborate molecular mechanisms and removes mutations arising from exposure to genotoxic and carcinogenic agents, including ROS [52]. These actions result from the presence of various factors and enzymes that impact on damaged DNA. Hence, DDR regulates important processes related to cell proliferation, cell cycle and induction of apoptosis [53]. The malfunctioning mechanism of response to DNA damage is closely related to cancerogenesis, as well as sensitivity to chemotherapy [54,55]. However, the excess of appearing abnormalities in DNA when the level of damage exceeds the repair capacity results in the initiation of the process of apoptosis, which is crucial in effective anticancer therapy [56].

DDR is initiated mainly by proteins from the phosphatidylinositol kinase family (PIKK), in particular ATM, ATR and DNA-PKcs leading to a series of further reactions [57]. In response to DNA damage, the cell cycle is temporarily arrested, and the error is processed before replication. DNA repair genes are stimulated by, i.e., ATM, ATR or H2AX, CHK1/2 [58,59]. These factors repair damages or begin apoptosis by enhancing the expression of p53 protein. As a consequence, the death-cell process occurs, inhibiting the overproliferation of abnormal cells [60]. In the present study, image cytometry analysis confirmed apoptotic cell death, but given the fact that the number of studies has shown that MCF-7 cells do not express caspase-3 [61,62], exact pathway must be confirmed. In studies on the cytotoxicity of phytochemicals, attention was paid to the role of caspase-2 in the signaling pathway initiated by DNA damage in liver and breast cancer [63,64]. Caspase-2 is the only one that is constitutively expressed in the cell nucleus. It is considered a component of DDR and can activate p-53-dependent apoptotic pathways as well as pathways that omit caspase-3, p53 and BCL-2. Caspase-2 can be activated independently by both DNA damage and oxidative stress [65]. The detection of apoptosis after the flavonoids, doxorubicin and combined treatment may indicate that this caspase could play an important role in each case.

As mentioned above, DOX, as well as flavonoids, is able to reduce the expression of genes involved in DNA repair. That could explain the lack of the cycle arrest and the DNA repair attempt under the influence of DOX and flavonoids in the concentrations used in the experiment. It has been confirmed in the present study that expression of all tested genes was downregulated by all tested compounds. Moreover, decrease in expression levels intensified during the simultaneous treatment with DOX and one of the flavonoids. The qPCR results showed that API + DOX reduced expression of over 70% of the genes: ERCC11, MSH2, MGMT and XPC. In turn, owing to using HESP + DOX, expression of ERCC1, ATM, OGG1 decreased by over 80%. Furthermore, the analysis of the results showed, that hesperidin co-administration with doxorubicin more strongly reduced every DNA repair gene expression (except for XPC) compared to the API + DOX sample. As a consequence, cells were unable to repair defects in the genetic material, which ultimately led to their death through the process of apoptosis. 

The question arises why, despite the intercalation and inhibition of replication enzymes, the expression of repair genes is significantly reduced. Considering the mechanism of action of doxorubicin, it can be assumed that the reason is a global inhibition of transcription. In addition to the fact that doxorubicin is a known topoisomerase inhibitor, Yang et al. revealed that DOX induced DSB at active gene promoters through torsion-based enhancement of nucleosome turnover [66]. In our previous studies related to the mechanism of action of doxorubicin, we have repeatedly observed a decrease in the expression of the examined genes under the influence of doxorubicin [39,67,68]. Synergistic action with flavonoids in this area confirms a similar mechanism of action—they interact with DNA, thus blocking the transcription process, their pro- and/or antioxidant effects seem to be less relevant. 

The observed mechanism of interaction between doxorubicin and apigenin/hesperidin corresponds to the therapeutic strategy of DDR inhibition [69,70]. In many cancers, the cells show defects in the DDR pathways. This makes them more susceptible to DNA damage and more dependent on other paths. The strategy includes the development of methods of treatment that focus on cancer-specific DDR dependencies. In this study, the inhibition of DNA repair was found but the effect seems not to be specific. The expression of all tested genes was decreased.

## 4. Materials and Methods 

### 4.1. Cell Culture and Treatment

The MCF-7 breast cancer line was used in this study (ATCC, USA). The cells were cultured in Dulbecco’s modified Eagle’s medium (DMEM) (Corning, Corning, USA) supplemented with 10% fetal bovine serum (Corning, Corning, USA), incubated at 37 °C with 5% CO2 in air atmosphere. The tested cells were treated for 48 h with 1 μM DOX (EBEWE Pharma, Unterach, Austria) and 50 μM of following HPLC standards (Sigma-Aldrich, USA): apigenin, hesperidin or combined (1 μM DOX + 50 μM single HPLC standard). The tested concentration of DOX was based on observed cytotoxicity for MCF-7 cell (IC50) and was consistent with plasma concentrations in patients treated with this drug. Doxorubicin is a standard drug used in breast cancer therapy [71]. Apigenin and hesperidin were used in optimal concentrations that sensitized the MCF-7 cells on DOX treatment in preliminary study.

### 4.2. MTT Assay

To determine cell viability, MTT assay was used. The test relied on the ability of living cells to reduce the orange tetrazolium salt (3-(4,5-dimethylthiazol-2-yl)-2,5-diphenyltetrazolium bromide) to water-insoluble purple formazan crystals. Therefore, the amount of formazan formed was proportional to the number of viable cells. The cells were seeded into 96-well plates in the concentration of 2 × 10^4^ cells/well. The tested compounds were added when 70–80% of confluence was achieved. After 48 h of incubation, the prepared MTT solution (0.5 mg/mL in phosphate buffered saline) was added to each well. After 4 h of incubation, MTT medium was removed and the crystals formed were dissolved in DMSO. The absorbance of the solution was measured at 570 nm with PowerWave XS microplate spectrophotometer (BioTek Instruments, Winooski, USA). Each assay was conducted three times and was measured in triplicates.

### 4.3. Assessment of Cells Morphology

Cell morphology was assessed after 48 h of incubation of cells with DOX-enriched medium and selected flavonoids by means of a phase-contrast microscope Nikon Eclipse Ti using NIS-Elements Imaging Software (Nikon, Tokyo, Japan).

### 4.4. Cell Cycle Analysis

The cell cycle was examined using NucleoCounter NC-3000 (ChemoMetec, Allerod, Denmark) according to the 2-step Cell Cycle Assay protocol (ChemoMetec, Allerod, Denmark). The cells were seeded into 6-well plates in the concentration of 4 × 10^5^ cells/well and the tested compounds were added when 70%–80% of confluence was achieved. After 48 h of incubation with tested compounds, the cells were moved out from the growing medium by suspension in 250 μL lysis buffer (Solution 10) enriched with 10 μg/mL DAPI (4′,6-Diamidine-2′-phenylindole dihydrochloride) and incubation was performed for 5 min at 37 °C. Then, 250 μL of stabilization buffer (Solution 11) was added. The cell suspension was loaded into the NC-Slide and read in NucleoCounter NC-3000. Each experiment was conducted three times with measurement in triplicate.

### 4.5. Apoptosis Detection

Detection of apoptosis was conducted using NucleoCounter NC-3000 (ChemoMetec, Allerod, Denmark) and the Annexin V Apoptosis Assay (ChemoMetec, Allerod, Denmark). The cells were seeded into 6-well plates in the concentration of 4 × 10^5^ cells/well and the tested compounds were added when 70%–80% of confluence was achieved. After a 48-h incubation, the cells were harvested using trypsin-EDTA solution (Corning, Corning, USA) and stained with the Annexin V–FITC (Fluorescein isothiocyanate), Hoechst 33342 and propidium iodide in compliance with the manufacturer’s recommended protocol. The cell suspension was loaded into the NC-Slide and read in NucleoCounter NC-3000. Each experiment was conducted three times with measurement in triplicate.

### 4.6. Reduced Glutathione (GSH) Level 

The GSH level in tested cells were evaluated with NucleoCounter NC-3000 (ChemoMetec, Allerod, Denmark) according to the manufacturer’s protocol (ChemoMetec, Allerod, Denmark). The cells were seeded into a 6-well plate at a concentration of 4 × 10^5^ cells/well and the tested compounds were added when 70–80% confluence was achieved. After a 48-h incubation, cells were harvested using trypsin-EDTA solution (Corning, Corning, USA). Next, the cells were stained with a solution of VitaBright-48™, which forms a strongly fluorescent product in the presence of GSH. The cell suspension was loaded into NC-Slide and read in NucleoCounter NC-3000. Each experiment was conducted three times with measurement in triplicate.

### 4.7. The Quantitative Real-Time PCR Analysis (qRT-PCR)

The cells were seeded into 6-well plates in the concentration of 4 × 10^5^ cells/well. The tested compounds were added when 70–80% of confluence was achieved. After 48 h of incubation, 1 mL of TRIzol™ Reagent (Invitrogen, Carlsbad, USA) was added to the culture dish to lyse the cells. Afterwards, the lysate was centrifuged for 5 min at 12,000× *g* at 4 °C, then the clear supernatant was processed according to the Chomczynski and Sacchi method [72]. Obtained RNA was reverse transcribed with an NG dART RT-PCR kit (EURx, Gdańsk, Poland) according to the manufacturer’s instructions. The qPCR was conducted using PowerUp SYBR Green Master Mix (ThermoFisher, Waltham, USA) according to manufacturer’s instructions in a 7500 Fast Real-Time PCR System (ThermoFisher, USA). The reaction was carried out in triplicates. The relative expression of tested genes was determined by qRT-PCR and the ΔΔCt method using *18SRNA* and *BACT* as reference genes. The statistical analysis was performed with RQ values (relative quantification, RQ = 2^−∆ΔCt).^ The primer sequences were summarized in the Table 3.

### 4.8. Determination of DNA Oxidative Damage

A DNA Damage Quantification Kit (Dojindo, Kumamoto, Japan) was used to evaluate oxidative DNA damage by measuring the quantity of AP sites, according to the manufacturer’s protocol. The cells were seeded into 6-well plates in a concentration of 4 × 10^5^ cells/well. The tested compounds were added when 70–80% of confluence was achieved. After 48-h incubation, the DNA was isolated with the Syngen DNA Mini Kit (Syngen, Wroclaw, Poland) according to the manufacturer’s protocol. In this method, an aldehyde reactive probe (ARP) reagent interacts specifically with an aldehyde group which is in the open ring form of the AP sites. After treating DNA with ARP reagent, AP sites were converted with biotin residues and were measured by avidin–biotin assay followed by a colorimetric detection of a horseradish peroxidase product at 650 nm using PowerWave™ microplate spectrophotometer (BioTek Instruments, Winooski, USA).

### 4.9. DNA Double-Strand Breaks (DSBs)

The determination of DSB was evaluated with a HCS DNA Damage Kit (Invitrogen, Carlsbad, USA) according to the manufacturer’s instruction. It is based on phosphorylated H2AX level measurement. The cells were seeded into 96-well plates in a concentration of 2 × 10^4^ cells/well. The tested compounds were added when 70–80% of confluence was achieved. After 48-h incubation, phosphorylated H2AX (Ser139), induced in response to double-strand breaks (DSBs) formation, was measured using specific primary antibody and Alexa Fluor™ 555 conjugated secondary antibody. The fluorescence signal was measured using the SpectraMax i3 Multi-Mode Platform (Molecular Devices, San Jose, USA).

### 4.10. Statistical Analysis

Statistical comparison of values was performed by one-way analysis of variance (ANOVA) and post-hoc multiple comparisons on a basis of Tukey’s honest significant difference test (Tukey’s HSD test) using STATISTICA 13 software (StatSoft, Krakow, Poland). All data were expressed as mean ± SD. Statistical significance was considered to be *p* < 0.05.

## 5. Conclusions

In conclusion, both apigenin and hesperidin enhance the cytotoxic effects of doxorubicin on breast cancer cells, and this phenomenon is accompanied by increased formation of DNA damage and a significant reduction in the expression of DNA repair genes. This may indicate a similar mechanism of action of the drug and flavonoids that is related to DNA disorders and consequently to replication and transcription stress. The pro- and antioxidant properties of all these compounds seem to be of lesser importance.

## Figures and Tables

**Figure 1 molecules-25-04421-f001:**
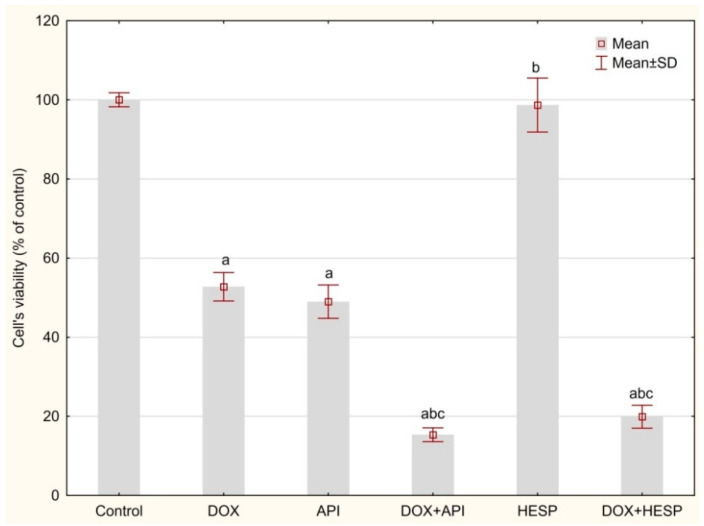
The MCF-7 cells viability (% of control) based on MTT assay. The cells were treated for 48 h with 1 µM of doxorubicin (DOX) and 50 μM of apigenin (API)/50 μM of hesperidin or combined (DOX + API, DOX + HESP). The values obtained from 3 independent experiments were presented as mean ± SD. a *p* < 0.05 vs. control, b *p* < 0.05 vs. DOX, c *p* < 0.05 vs. apigenin/hesperidin.

**Figure 2 molecules-25-04421-f002:**
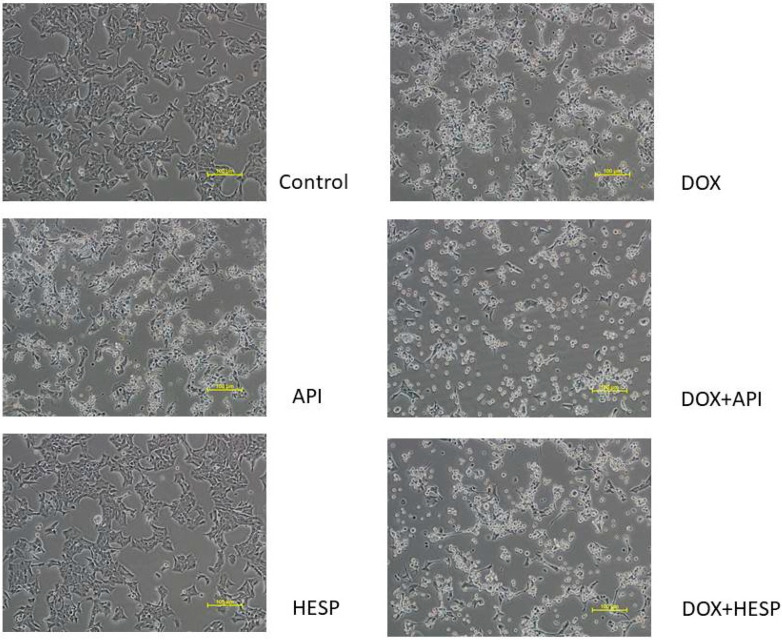
Morphological changes of MCF-7 cells. The cells were treated for 48 h with 1 µM of doxorubicin (DOX) and 50 μM of apigenin (API)/50 μM of hesperidin or combined (DOX + API, DOX + HESP). (Magnification × 200, scale bar = 100 μm)

**Figure 3 molecules-25-04421-f003:**
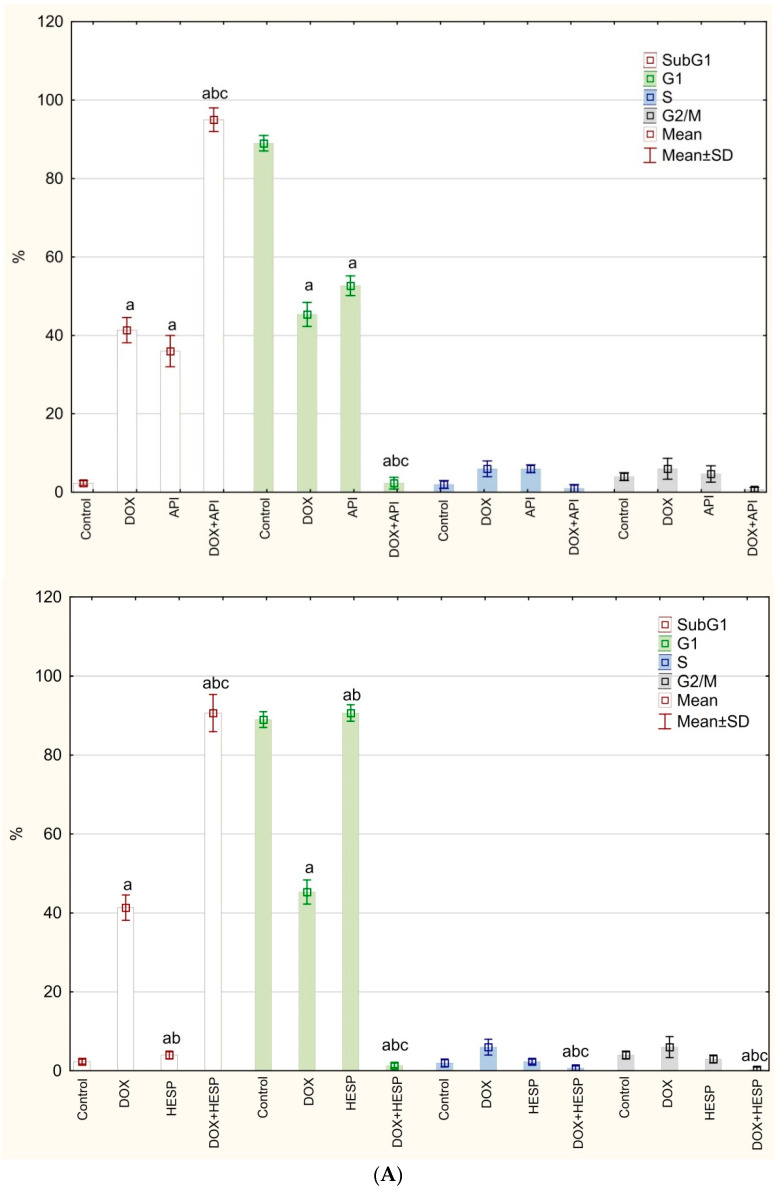
(**A**) Cell cycle analysis by image cytometry. The cells were treated for 48 h with 1 µM of doxorubicin (DOX) and 50 μM of apigenin (API)/50 μM of hesperidin or combined (DOX + API, DOX + HESP). The values obtained from 3 independent experiments were presented as mean ± SD. (**B**) Histograms representative of all repetitions of the experiment (M1—subG1, M2—G1, M3—S, M4—G2/M phase).

**Figure 4 molecules-25-04421-f004:**
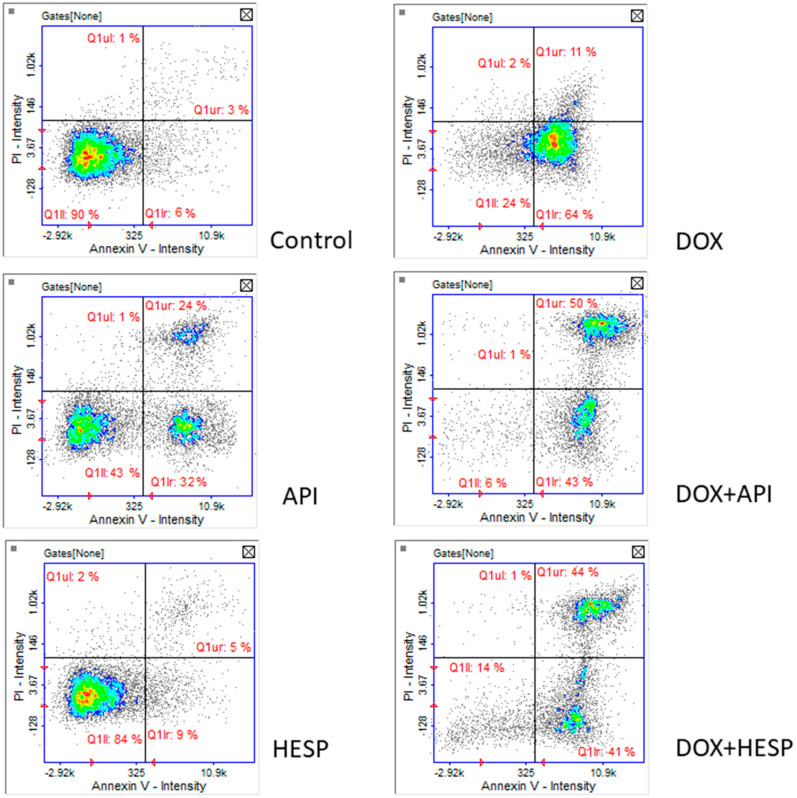
Cell apoptosis/necrosis detection in MCF-7 cells, stained with annexin V–FITC and propidium iodide for image cytometry. The cells were treated for 48 h with 1 µM of doxorubicin (DOX) and 50 μM of apigenin (API)/50 μM of hesperidin or combined (DOX + API, DOX + HESP). The results show one representative experiment of three independently performed. Q1II—live, Q1Ir—early apoptotic, Q1ur—late apoptotic and Q1uI—necrotic cells.

**Figure 5 molecules-25-04421-f005:**
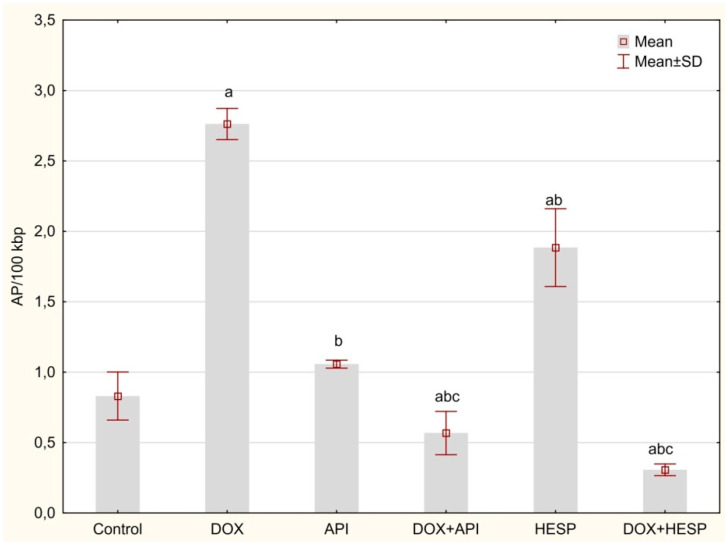
AP sites’ level in DNA of MCF-7 cells. The cells were treated for 48 h with 1 µM of doxorubicin (DOX) and 50 μM of apigenin (API)/50 μM of hesperidin or combined (DOX + API, DOX + HESP). The values obtained from 3 independent experiments were presented as mean ± SD. a *p* < 0.05 vs. control, b *p* < 0.05 vs. DOX, c *p* < 0.05 vs. apigenin/hesperidin.

**Figure 6 molecules-25-04421-f006:**
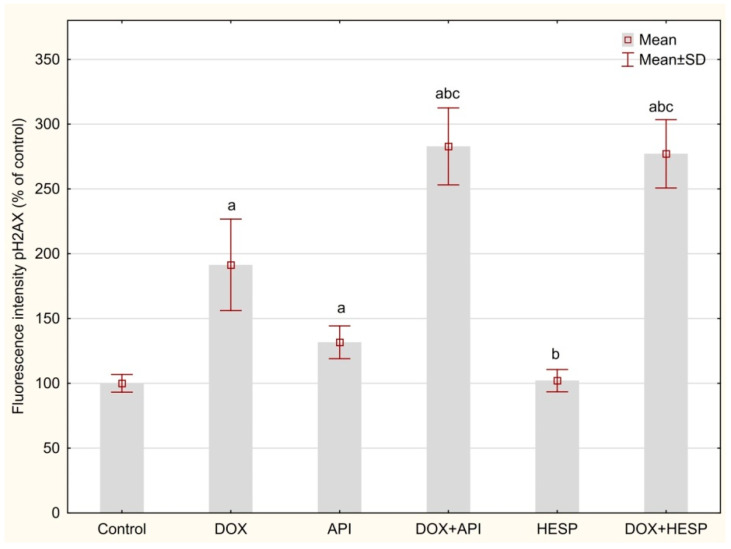
The content of double-strand breaks (DSBs) in DNA of MCF-7 cells (based on phosphorylated H2AX level) presented as a% of a control. The cells were treated for 48 h with 1 µM of doxorubicin (DOX) and 50 μM of apigenin (API)/50 μM of hesperidin or combined (DOX + API, DOX + HESP). The values obtained from 3 independent experiments were presented as mean ± SD. a *p* < 0.05 vs. control, b *p* < 0.05 vs. DOX, c *p* < 0.05 vs. apigenin/hesperidin.

**Figure 7 molecules-25-04421-f007:**
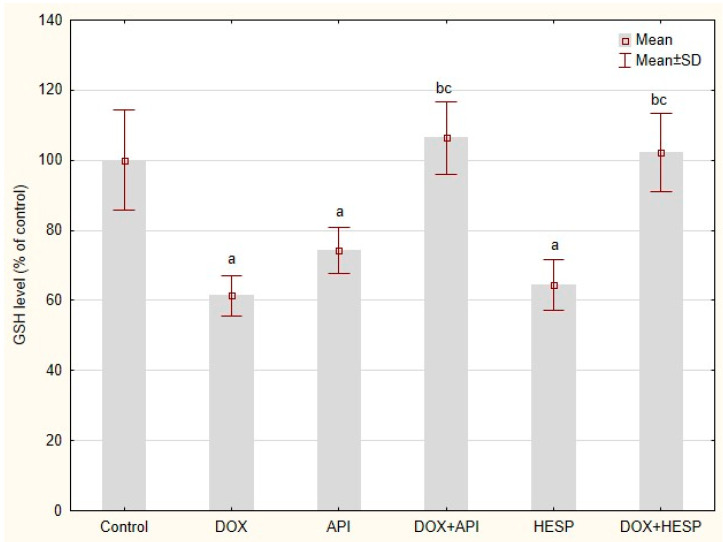
The GSH level in MCF-7 cells, presented as a % of a control. The cells were treated for 48 h with 1 µM of doxorubicin (DOX) and 50 μM of apigenin (API)/50 μM of hesperidin or combined (DOX + API, DOX + HESP). The values obtained from 3 independent experiments were presented as mean ± SD. a *p* < 0.05 vs. control, b *p* < 0.05 vs. DOX, c *p* < 0.05 vs. apigenin/hesperidin.

**Table 1 molecules-25-04421-t001:** The MCF-7 cells viability (% of control) based on MTT assay. The cells were treated for 48 h with 1 µM of doxorubicin (DOX) and apigenin (API) or hesperidin (HESP) in the range of concentrations 10–200 µM. The values obtained from 3 independent experiments were presented as mean ± SD. *^a^*
*p* < 0.05 vs. control, *^b^*
*p* < 0.05 vs. DOX, *^c^*
*p* < 0.05 vs. apigenin/hesperidin.

Control	DOX (1 µM)
100.04 ± 1.78	52.8 ± 3.59
**API (µM)**
10	25	50	100	200
95.67 ± 7.45 *^b^*	79.42 ± 6.11 *^ab^*	49.02 ± 4.23 *^a^*	33.07 ± 1.90 *^ab^*	12.77 ± 2.13 *^ab^*
**DOX + API**
49.04 ± 2.74 *^a^*	44.57 ± 9.02 *^ac^*	15.35 ± 2.35 *^abc^*	12.89 ± 3.07 *^abc^*	5.64 ± 1.15 *^abc^*
**HESP (µM)**
10	25	50	100	200
99.23 ± 7.73 *^b^*	101 ± 2.27 *^b^*	98.70 ± 6.82 *^b^*	75.886.67 *^ab^*	68.99 ± 3.57 *^ab^*
**DOX + HESP**
50.08 ± 4.86 *^ac^*	57.99 ± 2.31 *^ac^*	19.93 ± 2.80 *^abc^*	18.69 ± 4.55 *^abc^*	25.02 ± 4.54 *^abc^*

**Table 2 molecules-25-04421-t002:** Relative mRNA expression level of selected genes related to DNA repair. B-act and 18SN5 were used as reference genes. The results were calculated as RQ values and presented as mean ± SD value of three independent experiments. To compare more than two groups, the one-way analysis of variance ANOVA and the post-hoc multiple comparisons on the basis of Tukey’s HSD test were used. Significantly changed median RQ levels are marked with a color scale. DOX—1 μM doxorubicin, API—50 μM apigenin, HESP—50 μM hesperidin, DOX + API—1 μM doxorubicin and 50 μM apigenin, DOX + HESP—1 μM doxorubicin and 50 μM hesperidin. *^a^*
*p* < 0.05 vs. control, *^b^*
*p* < 0.05 vs. DOX, *^c^*
*p* < 0.05 vs. apigenin/hesperidin.

Table 1	Control		DOX		API		DOX + API		HESP		DOX + HESP		Scale (RQ)	
*PARP1*	1.000	0.023	1.057	0.057	0.816 *^ab^*	0.080	0.476 *^abc^*	0.020	0.391 *^ab^*	0.013	0.268 *^abc^*	0.008	> 2.00	
*ERCC1*	1.001	0.043	0.329 *^a^*	0.043	0.487 *^ab^*	0.027	0.208 *^abc^*	0.013	0.578 *^ab^*	0.019	0.126 *^abc^*	0.007	1.51–1.99	
*ATM*	1.051	0.428	0.234 *^a^*	0.043	0.622 *^ab^*	0.036	0.584 *^ab^*	0.014	0.715 *^ab^*	0.025	0.162 *^ac^*	0.048	1.11–1.50	
*MSH2*	1.002	0.085	0.369 *^a^*	0.017	0.235 *^ab^*	0.014	0.299 *^ab^*	0.007	0.765 *^ab^*	0.024	0.254 *^abc^*	0.033	0.91–1.10	
*OGG1*	1.000	0.031	0.785 *^a^*	0.042	0.766 *^a^*	0.042	0.425 *^abc^*	0.048	0.335 *^ab^*	0.008	0.177 *^abc^*	0.016	0.61–0.90	
*MGMT*	1.001	0.043	0.513 *^a^*	0.028	0.484 *^a^*	0.027	0.261 *^abc^*	0.031	0.664 *^ab^*	0.024	0.221 *^abc^*	0.012	0.21–0.6	
*XPC*	1.001	0.041	2.108 *^a^*	0.222	0.384 *^ab^*	0.016	0.145 *^abc^*	0.007	0.534 *^ab^*	0.023	1.190 *^abc^*	0.051	< 0.21	
*MLH1*	1.000	0.026	0.799 *^a^*	0.032	0.637 *^ab^*	0.043	0.387 *^abc^*	0.029	0.517 *^ab^*	0.008	0.357 *^abc^*	0.010		

**Table 3 molecules-25-04421-t003:** qPCR primers used in the experiment.

Target	Forward	Reverse
*PARP1*	CCCCACGACTTTGGGATGAA	AGACTGTAGGCCACCTCGAT
*ERCC1*	CTCGGAGTTTTGTGGGGGAC	CACTGGCGTCTACGTTCTCA
*ATM*	GCCGCGGTTGATACTACTTTG	GCAGCAGGGTGACAATAAACA
*MSH2*	CAGGAGGTGAGGAGGTTTCG	CCGTGCGCCGTATAGAAGTC
*MLH1*	GCACCGGGATCAGGAAAGAA	GCCTCACCTCGAAAGCCATA
*XPC*	GCGAAGTGGAATTTGCCCAG	TTGGCCTTGGATTTCTGGCT
*MGMT*	ACCGTTTGCGACTTGGTACT	TGCTCACAACCAGACAGCTC
*OGG1*	CCTGTGGGGACCTTATGCTG	TGTGAATCCCCTCTCCCGAT
*18SRNA*	GAAACTGCGAATGGCTCATTAAA	CACAGTTATCCAAGTGGGAGAGG
*BACT*	AGAGCTACGAGCTGCCTGAC	AGCACTGTGTTGGCGTACAG

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
