# Peer review of "Apigenin and Hesperidin Downregulate DNA Repair Genes in MCF-7 Breast Cancer Cells and Augment Doxorubicin Toxicity"

_molecules, 2020, doi:10.3390/molecules25194421_

Round 1

Reviewer 1 Report

The study by Korga-Plewko et al. (Apigenin and hesperidin downregulate DNA repair genes in MCF-7 breast cancer cells and augment doxorubicin toxicity) evaluated the oncostatic effects of apigenin and hesperidin as single agents and in combination with doxorubicin in vitro. Authors performed an analysis of cytotoxicity, cell cycle, oxidative damage, DNA double-strand breaks, and expression of genes associated with DNA repair. Results showed that both flavonoids enhanced the cytotoxic effect of doxorubicin (evaluated by MTT test) without significant inhibition of cell cycle. The results of DNA damage analysis were not consistent. Hesperidin, but not apigenin, induced DNA oxidative damage, similarly as doxorubicin but the combined therapy was not effective. On the other hand, both flavonoids potentiated the effect of doxorubicin regarding the rate of DNA double-strand breaks formation. Analysis of glutathione level indicated the prooxidant effects of all three agents in a single therapy, but, surprisingly, the combination of apigenin/hesperidin with doxorubicin did not alter glutathione levels. Assessment of DNA repair genes expression showed downregulation of the most of evaluated genes, both in single and combined therapy. The results suggest that both substances may potentiate the oncostatic efficacy of doxorubicin, however, this has to be evaluated in vivo.

This is a well-designed study, the conclusions are clear and are adequately supported by results. However, the authors included only a single cell line in their analysis. The quality of the study would have been enhanced if authors had also used other lines, particularly TNBC (triple-negative breast cancer) line, as the options to treat TNBC are limited, and therefore there is intense interest in finding new medications or supporting therapies for this kind of neoplasia. Another weak point is that some aspects of the present study have been addressed and reported before. For example, augmentation of growth inhibitory effects of doxorubicin by apigenin has been reported in 2007 by Brantley et al., induction of DNA damage and oxidative stress by various concentrations of apigenin in MCF-7 and also MDA-MB-231 cells has been evaluated by Madunic et al. in 2018; cytotoxic effect of hesperidin in MCF-7 cells resistant to doxorubicin has been reported by Febriansah, 2014.

I would like to make some comments and suggestions.

  1. The Introduction part is rather vague. The beneficial effects of flavonoids are well-known but flavonoids represent a large and heterogeneous group of substances and the authors should have focused more on flavonoids they used in their study. It may be of interest for readers to include more information on anticancer effects of both flavonoids (including antiangiogenic and immunomodulatory properties) and specify the cancer type, plus to mention relevant in vivo
  2. Line 79-81: the authors mentioned preliminary studies regarding the optimal dose of doxorubicin, it may be of interest for readers to add a reference in case those results were published.
  3. Line 117-119: Two graphs in Figure 3A are identical.
  4. Line 219-225: The antioxidant activity of the combination of doxorubicin and hesperidin/apigenin found by authors is unexpected as single agents (with the exception of single apigenin) showed prooxidant effects. Is there any explanation for this phenomenon, respectively, was this reported before?
  5. Line 283-284­: “The tested concentration of DOX was based on observed cytotoxicity for MCF-7 cell 283 (IC50) and was consistent with plasma concentrations in patients treated with this drug.” Please provide a reference.
  6. Line 27, 30, 81-2, 228, 360: In regards to the toxic effects of doxorubicin, I believe that the more suitable term would be “cytotoxic”.

Are there any studies on the possible toxicity of apigenin and hesperidin on normal cells? This has to be considered in the supportive treatment of cancer.

Author Response

Response to Reviewer 1 Comments

We sincerely thank for thorough reading of the manuscript, your valuable comments and constructive suggestions which helped us with improving the quality of our manuscript. Accordingly, the revised manuscript has been systematically improved. Our responses to the comments are given below.

  • …the authors included only a single cell line in their analysis. The quality of the study would have been enhanced if authors had also used other lines, particularly TNBC (triple-negative breast cancer) line, as the options to treat TNBC are limited, and therefore there is intense interest in finding new medications or supporting therapies for this kind of neoplasia

Thank you very much for that comment. If we consider the use of apigenin or hesperidin in the treatment of breast cancer, it is extremely important to characterise the response to treatment in relation to the molecular type of cancer. There are a number of reports that have investigated such relation [as reviewed by Shukla and Gupta Pharm Res. 2010]. The aim of our study was to explain whether the known mechanism of intensified DOX cytotoxicity by apigenin/hesperidin is associated with oxidative or non-oxidative DNA damage and what is the contribution of DDR to this phenomenon. This is why we have chosen an MCF-7 line with a proven sensitivity to both DOX and the flavonoids studied

  • …Another weak point is that some aspects of the present study have been addressed and reported before. For example, augmentation of growth inhibitory effects of doxorubicin by apigenin has been reported in 2007 by Brantley et al., induction of DNA damage and oxidative stress by various concentrations of apigenin in MCF-7 and also MDA-MB-231 cells has been evaluated by Madunic et al. in 2018; cytotoxic effect of hesperidin in MCF-7 cells resistant to doxorubicin has been reported by Febriansah, 2014.

It is defenitely true that apigenin and hesperidin have long been studied as therapeutic factors in breast cancer and their ability to sensitize cancer cells to doxorubicin has been confirmed. We point this out in the introduction. The fact is that this phenomenon is not fully elucidated. Additionally, both DOX and flavonoids influence red-ox status of cells and affect DNA. The numerous possible mechanisms of action make the observed effect of simultanous treatment not obvious. That is why our research has focused on the mechanism of synergy - not on the existence of synergies itself.

  • The Introduction part is rather vague. The beneficial effects of flavonoids are well-known but flavonoids represent a large and heterogeneous group of substances and the authors should have focused more on flavonoids they used in their study. It may be of interest for readers to include more information on anticancer effects of both flavonoids (including antiangiogenic and immunomodulatory properties) and specify the cancer type, plus to mention relevant in vivo

We agree with the reviewer on this point. The introduction has been corrected . General statements on flavonoids have been supplemented with specific data concerning apigenin and hesperidin. However, we focused on the information necessary to understand the purpose of the work. The available data on the anticancer effects of apigenin and hesperidin are too extensive to be included in the introduction section.

  • Line 79-81: the authors mentioned preliminary studies regarding the optimal dose of doxorubicin, it may be of interest for readers to add a reference in case those results were published.

The table with preliminary study result was included in the manuscript (Result section)

  • Line 117-119: Two graphs in Figure 3A are identical

Thank you for your caution. The duplicate graph has been replaced by a proper one.

  • Line 219-225: The antioxidant activity of the combination of doxorubicin and hesperidin/apigenin found by authors is unexpected as single agents (with the exception of single apigenin) showed prooxidant effects. Is there any explanation for this phenomenon, respectively, was this reported before

As we mentioned above, the known, possible mechanisms of action of DOX and flavonoid make the effect of simultanous treatment not obvious. We tried to comment unexpected results in discussion section (line 283 -293), however there is no enough data to fully explain this phenomenon.

  • Line 283-284¬: “The tested concentration of DOX was based on observed cytotoxicity for MCF-7 cell 283 (IC50) and was consistent with plasma concentrations in patients treated with this drug.” Please provide a reference.

The reference was provided (no 67)

  • Line 27, 30, 81-2, 228, 360: In regards to the toxic effects of doxorubicin, I believe that the more suitable term would be “cytotoxic”.

We agree with the reviewer’s statement. The term was corrected at the indicated locations

  • Are there any studies on the possible toxicity of apigenin and hesperidin on normal cells? This has to be considered in the supportive treatment of cancer.

We agree that this is a key issue. There are reports that apigenin nor hesperidin are not toxic for normal cells and we included apriopriate information in introduction section (line 77-78)

Reviewer 2 Report

The work by Agnieszka Korga-Plewko et al. (ID: Molecules 928558) is interesting and investigates the antitumor properties of a combination of the conventional anticancer drug doxorubicin with the flavonoids apigenin and hesperidin against the MCF-7 breast cancer cells. Interestingly, it seems that the biological properties of apigenin are linked to the induction of DNA double strand breaks, while the hesperidin is able to increase the DNA oxidative damage. Some Major and Minor revisions are required to improve this Manuscript.

MAJOR REVISIONS

-In Figure 3, the authors show that “DOX”, “API”, “DOX+API” and “DOX+HESP” treatments remarkably increased the % MCF-7 cells in Sub-G1 phase, while the “HESP” treatment had no effect. Since the increase of % cells in Sub-G1 phase is linked to cell death and activation of the apoptotic process, the authors should evaluate if the treatments “DOX”, “API”, “DOX+API”, “HESP” and “DOX+HESP” increase the activity levels of the effector Caspase 3 in MCF-7 breast cancer cells.

-In Figure 4, the authors showed that the treatments “DOX” and “HESP” increased the DNA oxidative damage in MCF-7 tumor cells, while Figure 5 shows that the treatments “DOX”, “API”, “DOX+API”, “DOX+HESP” increased the double strand DNA breaks in MCF7 breast cancer cells. Since the initiator Caspase 2 is activated by various DNA damages [as described in Antonini E. et al., Nutr. Cancer, 2018], the authors should evaluate if the treatments “DOX”, “API”, “DOX+API”, “HESP” and “DOX+HESP” induce an activation of caspase 2 in MCF-7 breast cancer cells. The new results regarding caspase 3 and caspase 2 could significantly improve the Manuscript.

MINOR REVISIONS

-In Figure 3, I noticed that there is a duplicate of Fig. 3A: is it necessary or can it be removed?

-In Table 1, please add the statistical analysis also to the column “HESP” for the modulation of the expression levels of PARP1, ERCC1, ATM, MSH2, OGG1, MGMT, XPC and MLH1 genes.

-In the Table 1, how did the authors evaluate the Standard Deviation of the expression levels of PARP1, ERCC1, ATM, MSH2, OGG1, MGMT, XPC and MLH1 in control samples? Because the authors arbitrarily set the expression values to 1 for RTqPCR, which usually implies a SD value of 0 for control samples.

-Furthermore, there are some typos that should be corrected:

1)Abstract: hesperidin instead of hesperidine.

2)Introduction: studies have established instead of studies have been established

3)Introduction: there is an increasing number of researches instead of there is numerous researches

4)Introduction: unable to repair DNA, leading to pro-apoptotic effects instead of unable to DNA repair, hence they inducing pro-apoptotic effects.

5)Materials and Methods: in the concentration of 2x104 cells/well instead of in the concentration of 2x104 cells/well.

6)Materials and Methods: the clear supernatant was processed instead of the clear supernatant was proceed

7)Results: 15.35±1.75% residual viability instead of 15.35±1.75% viability. 

8)Results: The determination of oxidative DNA damage evidenced that DOX instead of The determination of oxidative DNA damage presented that DOX

9)Results: this gene expression levels compared to control instead of this gene expression levels compering to control.

10)Discussion: Genotoxic effects were associated instead of Genotoxic effects was associated

11)Discussion: occurred by reducing oxidative damage instead of occurred by reducing oxidative

12)Discussion: we observed an increase of cells in subG1 phase instead of there were subG1 phase observed

 13)Discussion: inhibiting the over proliferation of abnormal cells instead of and the abnormal cells, instead of over proliferating.

14)Discussion: Moreover, decrease of expression levels intensified instead of Moreover, lessen of expression intensified

15)Conclusions: of all these compounds instead of of these all compounds. 

Author Response

Response to Reviewer 2 Comments

We sincerely thank for thorough reading of the manuscript, your valuable comments and constructive suggestions which helped us with improving the quality of our manuscript. Accordingly, the revised manuscript has been systematically improved. Our responses to the comments are given below.

1) n Figure 3, the authors show that “DOX”, “API”, “DOX+API” and “DOX+HESP” treatments remarkably increased the % MCF-7 cells in Sub-G1 phase, while the “HESP” treatment had no effect. Since the increase of % cells in Sub-G1 phase is linked to cell death and activation of the apoptotic process, the authors should evaluate if the treatments “DOX”, “API”, “DOX+API”, “HESP” and “DOX+HESP” increase the activity levels of the effector Caspase 3 in MCF-7 breast cancer cells.

We agree that analysis of the cell cycle and revealed the presence of the subG1 phase is not a confirmation of apoptotic cell death. Since a number of studies have previously shown that MCF-7 cells do not express caspase-3 (Kagawa 2001Cancer Biology; Jänicke 2009Breast Cancer Res Treat. ; Wang 2016 J Breast Cancer.) we have attached the results of apoptotsis/necrosis detection using image cytometry and annexin V–FITC and propidium iodide staining.

  • In Figure 4, the authors showed that the treatments “DOX” and “HESP” increased the DNA oxidative damage in MCF-7 tumor cells, while Figure 5 shows that the treatments “DOX”, “API”, “DOX+API”, “DOX+HESP” increased the double strand DNA breaks in MCF7 breast cancer cells. Since the initiator Caspase 2 is activated by various DNA damages [as described in Antonini E. et al., Nutr. Cancer, 2018], the authors should evaluate if the treatments “DOX”, “API”, “DOX+API”, “HESP” and “DOX+HESP” induce an activation of caspase 2 in MCF-7 breast cancer cells. The new results regarding caspase 3 and caspase 2 could significantly improve the Manuscript

Caspase-2 activity determination is an interesting and valuable parameter related to DNA damage. I have to admit that we did not take it into account in our research before. Due to the short time available to correct the manuscript, we did not have the technical possibility to carry out this analysis. However, after reviewing the available reports, we will certainly consider this in the future.

  • In Figure 3, I noticed that there is a duplicate of Fig. 3A: is it necessary or can it be removed?

Thank you for your caution. The duplicate graph has been replaced by a proper one.

  • In Table 1, please add the statistical analysis also to the column “HESP” for the modulation of the expression levels of PARP1, ERCC1, ATM, MSH2, OGG1, MGMT, XPC and MLH1 genes.

Statistical analysis results were added

  • In the Table 1, how did the authors evaluate the Standard Deviation of the expression levels of PARP1, ERCC1, ATM, MSH2, OGG1, MGMT, XPC and MLH1 in control samples? Because the authors arbitrarily set the expression values to 1 for RTqPCR, which usually implies a SD value of 0 for control samples.

Thank you for your caution. This is an error that occurred while copying data into a table – it was corrected.

  • …Furthermore, there are some typos that should be corrected:
  • Abstract: hesperidin instead of hesperidine – corrected (line 30)
  • Introduction: studies have established instead of studies have been established - corrected (line 39)
  • Introduction: there is an increasing number of researches instead of there is numerous researches - corrected (line 46)
  • Introduction: unable to repair DNA, leading to pro-apoptotic effects instead of unable to DNA repair, hence they inducing pro-apoptotic effects. - corrected (line 67)
  • Materials and Methods: in the concentration of 2x104 cells/well instead of in the concentration of 2x104 cells/well. – corrected in all method sections
  • Materials and Methods: the clear supernatant was processed instead of the clear supernatant was proceed - corrected (line 406)
  • Results: 35±1.75% residual viability instead of 15.35±1.75% viability.  - corrected (line 96)
  • Results: The determination of oxidative DNA damage evidenced that DOX instead of The determination of oxidative DNA damage presented that DOX - corrected (line 177)
  • Results: this gene expression levels compared to control instead of this gene expression levels compering to control. - corrected (line 230)
  • Discussion: Genotoxic effects were associated instead of Genotoxic effects was associated - corrected (line 243)
  • Discussion: occurred by reducing oxidative damage instead of occurred by reducing oxidative - corrected (line 253)
  • Discussion: we observed an increase of cells in subG1 phase instead of there were subG1 phase observed - corrected (line 272)
  • Discussion: inhibiting the over proliferation of abnormal cells instead of and the abnormal cells, instead of over proliferating. - corrected (line 314)
  • Discussion: Moreover, decrease of expression levels intensified instead of Moreover, lessen of expression intensified - corrected (line 319)
  • Conclusions: of all these compounds instead of of these all compounds.  - corrected (line 452)

Round 2

Reviewer 1 Report

I have no further comments.

Author Response

Dear Reviewer,

We sincerely thank you for your cooperation.

Reviewer 2 Report

Agnieszka Korga-Plewko et al. answered to almost all my questions and the data obtained through the Annexin V assays really improved their Manuscript. I think that one last minor revision, regarding the role of caspase 2, is required. In fact, Yuet Ping Kwan et al. [Pharm. Biol., 2016] published a very interesting Paper regarding the pro-apoptotic role of Euphorbia hirta L against MCF-7 breast cancer cells; these authors showed that the phytochemicals of this natural extract activated caspases 2,8,9 and 6 in MCF-7 tumor cells. The authors should discuss the results of Yuet Ping Kwan et al. in order to describe the role of caspase 2 and ROS in the antiproliferative effects of phytochemicals against breast cancer cells. 

Author Response

Dear Reviewer,

We sincerely thank for your comments. In Discussion section (line 270 - 279) we tried to identify the possible role of caspase-2 in cytotoxicity observed in present study. However, since we did not study caspase-2 activity, we tried to avoid over-discussing this topic. We hope you find it sufficient.